# X−ECG: Explainable Foundation model for Electrocardiogram

## Abstract

Electrocardiography (ECG) is a cost-effective and widely accessible tool for evaluating cardiac health. While numerous machine learning methods have been developed to assist cardiologists in diagnosis, many suffer from lacking explainability, making it difficult to understand why a particular disease is classified. To address this limitation, we introduce X-ECG, an explainable ECG foundation model. To train this model, we first curate wave-level anomalies annotations on public datasets, using a rule-based algorithm that finds abnormal waves, intervals or segments in ECG signal according to established clinical knowledge. To help models learn where to focus, we propose an attention-guided training approach that enables the model to highlight relevant regions. To the best of our knowledge, X-ECG is the first ECG foundation model with built-in explainability. Our experiments show that using our dataset to guide the model not only adds explainability but also improves performance in arrhythmia classification and report generation tasks.

## 1 Introduction

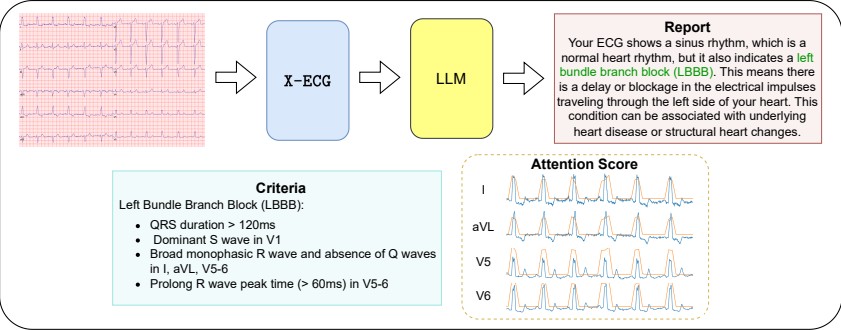

Figure 1: X-ECG is capable of identifying abnormal regions that contribute to its output, thereby enhancing model interpretability. This is achieved by guiding its attention scores during training to focus on these critical regions.

The electrocardiogram (ECG) is a monitoring tool that provides a window into the heart's electrical activity and offers valuable insights into cardiac health. To interpret an ECG sample, cardiologists typically follow a predefined process, examining specific regions and assessing abnormalities. A full and detailed interpretation can take up to five minutes, depending on the cardiologist's level of experience Bortolotti et al. (2025).

To accelerate and support this process, numerous machine learning methods have been developed. Many groups have proposed classification models with excellent performance across different benchmarks Strodthoff et al. (2020); Kiranyaz et al. (2016); Wang et al. (2021); Herman et al. (2024), while others have focused on building foundation models that can be adapted to a wide range of downstream tasks Song et al. (2025); McKeen et al. (2025); Wang et al. (2025); Jin et al. (2025); Nguyen et al. (2025). Notably, McKeen et al. (2025) and Nguyen et al. (2025) extend this progress by addressing challenges in interpreting corrupted ECG signals.

However, to fully aid doctors in performing a diagnosis, instead of just returning the final disease class, it is desirable for the model to highlight the regions that contribute to such decision making, as it helps the doctors to understand the decision. While anomaly detection and localization models like Bui et al. (2024); Jiang et al. (2023b; 2024) can do exactly that by highlighting irregular beats, they do not provide diagnosis from those anomalies.

Existing ECG datasets, most notably MIMIC-IV-ECG and PTB-XL, typically have the waveform and metadata containing the disease class(es) associated with each waveform. These datasets however do not have locations of the anomalies on the waveforms. Models are typically trained following a black-box training approach on these massive dataset with the waveform as input and class label as outputs, leading to models with lack of explainability. Therefore, fine-grained annotations of anomalies associated with each disease class is crucial for building explainable ECG systems.

In this study, we curate an algorithm that segments anomalous waves, segments, and intervals that are associated with each cardiac disease, based on an established clinical knowledge database. Specifically, we design a two-stage procedure which first finds the abnormal waves, intervals and segments for each lead in the ECG signal, and then to further finetune the annotations, we employ the Criteria Feature Retrieval (CFR) proposed in Nguyen et al. (2025) to only highlight the abnormal components contribute to the final report. We apply this algorithm on MIMIC-IV-ECG and PTB-XL, creating fine-grained, wave-level anomaly annotations, which we will refer to as `MIMIC-IV-ECG+X` and `PTB-XL+X` respectively. We then propose a training framework guided by such heatmaps from those dataset to build `X-ECG`, an ECG foundation model with explainability. To showcase its utility, we augment `X-ECG` with a classification module for cardiac diagnosis and an LLM for automated report generation. By utilizing this dataset in guiding the attention mechanism of the ECG encoder, the model not only have a boost in arrhythmia classification and report generation performance, but also can show the location of anomalies contribute to the final decision.

The main contributions of the paper is summarized as below:

- We construct `MIMIC-IV-ECG+X` and `PTB-XL+X`, a dataset for training and benchmarking the model ability to segment abnormal locations respectively.
- To help the model in efficiently learning the correct attention location, we employ a Attention-guided mechanism that utilizes that heatmap from `X-ECG` and model attention weights.
- We demonstrate that the guided model have a boost performance in various tasks, such as arrhythmia classification, report generation and anomaly localization.

## 2 RELATED WORK

### 2.1 FOUNDATION MODELS FOR ECG

In recent years, foundation models have emerged as a transformative paradigm across various domains, including ECG analysis. These models are typically pretrained on large-scale datasets and subsequently adapted to a wide range of downstream tasks through transfer learning or fine-tuning. To build such models, researchers often adopt pretraining frameworks like multimodal alignment or self-supervised learninig.

METS Li et al. (2023) and MERL Liu et al. (2024) employ the dual-alignment strategy to align ECG signals with their corresponding text reports, enabling the learning of high-quality ECG representations. Building upon METS, MERL introduces an inter-alignment module within the signal modality to further refine these representations. By leveraging these techniques and training on large-scale datasets, both METS and MERL demonstrate the capability for zero-shot cardiac classification—allowing the identification of previously unseen diseases without the need for additional labeled data.

Similarly, ESI Yu et al. (2024), ECG-Chat Zhao et al. (2025) and TolerantECG Nguyen et al. (2025) also adopt a multimodal alignment pretraining framework. Due to the presence of samples with highly similar text reports, training noise can arise and hinder representation learning. To mitigate this issue, ESI incorporates a retrieval-augmented generation pipeline, TolerantECG applies a more straightforward feature retrieval method, and ECG-Chat leverages signal interval durations to en-

hance report diversity. By aligning ECG signals with semantically enriched reports, both models demonstrate improved performance on downstream tasks such as arrhythmia diagnosis and ECG-report retrieval.

In contrast to the aforementioned methods, ST-MEM Na et al. (2024) adopts a self-supervised learning strategy focused on reconstructing masked ECG patches. By training the model to recover the full 12-lead ECG signal, ST-MEM effectively learns joint spatio-temporal representations. This approach not only yields strong performance in arrhythmia classification, but also demonstrates adaptability to varying lead configurations—highlighting the spatial awareness embedded in its learned representations.

ECG-FM McKeen et al. (2025) is built upon a transformer-based design. The training process incorporates domain-specific augmentations tailored to ECG data, along with a dual-objective strategy combining contrastive learning and signal masking. This setup allows the model to capture rich contextual representations, enhancing its ability to generalize across multiple tasks—including identifying specific cardiac conditions from ECG readings, detecting reduced left ventricular ejection fraction, and recognizing abnormal levels of cardiac troponin (cTn).

## 2.2 EXPLAINABLE AI

Explainable AI has become a vital area of research, aiming to enhance the transparency and interpretability of complex machine learning models, particularly deep learning systems. This need is especially pronounced in the medical domain, where trust in model outputs is often limited due to the lack of clear explanations behind predictions.

Among the most widely adopted techniques for visualizing model decision-making is Grad-CAM Selvaraju et al. (2019), which produces heatmaps that highlight influential input features—such as image regions—that contribute to a model's output. Grad-CAM and its variants have been particularly impactful in computer vision, offering intuitive visual explanations for convolutional neural networks.

For Transformer-based architectures, a complementary technique is attention calibration Lu et al. (2021); Zhou et al. (2024). This method introduces an auxiliary loss function during training, guided by predefined attention heatmap labels. By shaping the model's attention distribution, it helps determine the appropriate level of focus each token should receive, thereby improving interpretability and aligning attention with human-understandable patterns.

## 3 DATA CURATION

To support both the generation of `MIMIC-IV-ECG+X` and `PTB-XL+X` with clinical expertise and the evaluation of final outputs, we collaborated with medical students and doctors.

Despite unprecedented advancement in AI for healthcare domain, physicians are cautious about adopting healtcare AI models that lack explainability because opaque predictions undermine trust and accountability in clinical decision-making. Without clear reasoning, clinicians cannot assess whether outputs are based on sound medical evidence or spurious patterns, raising concerns about patient safety and liability. Additionally, these AI models are usually trained using as a blackbox with a pre-defined dataset. Since data is required for efficient models, to make them interpretable, we generate `MIMIC-IV-ECG+X` and `PTB-XL+X` that equip with abnormal heatmap region for each ECG signal. This dataset requires two-stage rule-based procedure: *Abnormal seeking* and *Criteria seeking*

### 3.1 ABNORMAL SEEKING

We first identify the locations that represent irregular features. Specifically, we gather all the conditions of abnormal representations across the *Life in the Fastlane* ECG library lit (2008). From those features, we can find which waves, intervals or segments that need to pay high attention to and highlight them. To enable this algorithm, the input ECG signal must first be segmented into three fundamental components: the P wave, QRS complex, and T wave. We achieve this by training a UNet3+ model Huang et al. (2020) on the LuDB dataset Kalyakulina et al. (2020), following

Table 1: Abnormal conditions for each component in ECG signal

| Components | Abnormal conditions |
|------------|---------------------|
| P wave | • Missing
• Amplitude > 0.25mV in limb leads
• Amplitude > 0.15mV in precordial leads
• Duration > 120ms
• Inverted in lead I, II, III
• Upright in lead aVR |
| QRS complex | • Duration > 120ms
• Have odd shape (RSR', QR, rS) |
| T wave | • Amplitude > 0.5mV in limb leads
• Amplitude > 1mV in precordial leads
• Upright in all leads except aVR and V1 |
| PR interval | • Duration > 200ms
• Duration < 120ms |
| QT interval | • Corrected duration > 440ms
• Corrected duration < 350ms |
| ST segment | • Elevate: > 0.1mV isoelectric line (> 0.05mV in lead V2 and V3)
• Depression: < 0.5mV isoelectric line |

the methodology outlined by Joung et al. (2024). Once segmented, we compute key interval features—namely PR intervals, QT intervals, and ST segments—for each heartbeat. Abnormal regions are then identified based on the criteria specified in Table 1. For QT interval analysis, we apply Bazett's correction to account for heart rate variability prior to detecting abnormalities:

$$Q_i T_i^{correct} = \frac{Q_i T_i}{\sqrt{R_{i-1} R_i}},$$

where $Q_i T_i$ is the QT duration in the $i$-th rhythm, and $R_{i-1} R_i$ is the duration from R peak in the $(i-1)$-th rhythm to R peak in the $i$-th rhythm.

### 3.2 CRITERIA SEEKING

After identifying the initial abnormal locations, we refine them using the CFR module Nguyen et al. (2025), as illustrated in Figure 2. This module provides diagnostic criteria that specify which ECG components contribute to each diagnosis, enabling the removal of redundant or noise-induced false positives. For each diagnosis in the dataset, we retrieve its associated criteria and use the specified abnormal components to extract the relevant segments identified earlier. In cases where certain components were missed during the initial segmentation, we still highlight them if they are referenced in the diagnostic criteria. For further details on the generated heatmap dataset, please refer to Appendix A.

## 4 MODEL ARCHITECTURE

To showcase the advantages of our synthesized X-ECG dataset in arrhythmia classification, anomaly localization, and report generation, we train a foundational model named X-ECG, as illustrated in Figure 3. The model is built upon the CoCa framework Yu et al. (2022), which has demonstrated strong performance on ECG datasets Yu et al. (2024); Zhao et al. (2025). To enrich the model with spatial and temporal context, we incorporate the Spatial-Temporal embedding proposed by Jin et al. (2025), which encodes lead-specific and time-frame information for each token. Our experiments reveal that this embedding not only enhances classification accuracy but also significantly improves the model's ability to localize abnormal regions within the ECG signal.

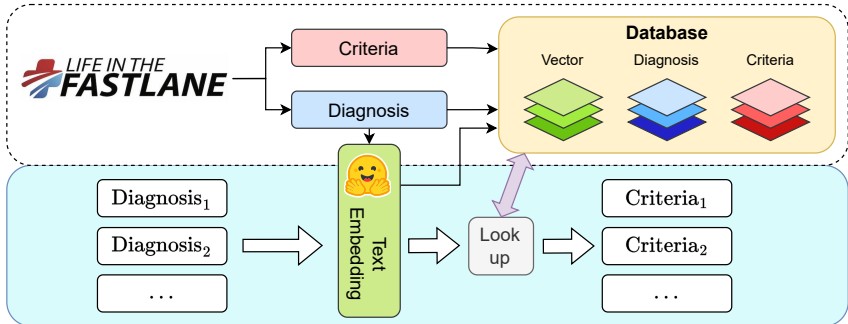

Figure 2: The Criteria Feature Retrieval (CFR) module. It returns diagnostic criteria for each input diagnosis by comparing its embedding with those stored in a pre-constructed vector database.

## 4.1 ECG SIGNAL AND TEXT REPORT ALIGNMENT

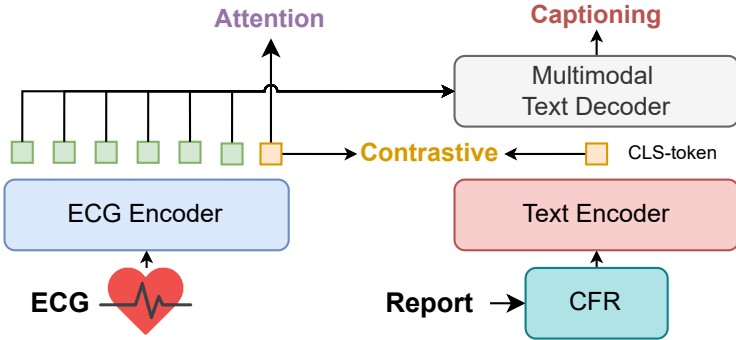

Figure 3: Overview of the X-ECG architecture. It comprises an ECG encoder and a text encoder that process the input ECG waveforms and their corresponding comprehensive augmented reports, respectively. Contrastive learning is performed using the CLS tokens from both encoders to align cross-modal representations. A multimodal text decoder is further employed to reconstruct the input report, conditioned on the ECG token embeddings. To guide focus regions, attention scores between the CLS token and other ECG tokens are optimized using an attention-specific loss function.

Following the previous ECG language pretraining approach Li et al. (2023); Liu et al. (2024); Yu et al. (2024); Nguyen et al. (2025), as shown in Figure 3, we first employ two pretraining objectives for comprehensive learning, including contrastive loss for robust representation learning and captioning loss for semantic alignment. Both encoders aim to project the inputting ECG and text into a unified embedding space. Consequently, the ECG encoder and the Multimodal Text Decoder are jointly optimized by contrasting the paired text against others in the sampled batch, while the Text Encoder is frozen with pretrained weights.

Optimizing the contrastive loss enables alignment between ECG features $\mathbf{E}$ and corresponding text reports $\mathbf{T}$. This process is illustrated across a batch $B$ of training samples:

$$\mathcal{L}_{contrastive} = -\frac{1}{2}\left(\underbrace{\sum_{i=1}^{B}\log\frac{\exp(\mathbf{E}_i^\top\mathbf{T}_i/\tau)}{\sum_{j=1}^{B}\exp(\mathbf{E}_i^\top\mathbf{T}_j/\tau)}}_{\text{ecg-to-text}} + \underbrace{\sum_{i=1}^{B}\log\frac{\exp(\mathbf{T}_i^\top\mathbf{E}_i/\tau)}{\sum_{j=1}^{B}\exp(\mathbf{T}_i^\top\mathbf{E}_j/\tau)}}_{\text{text-to-ecg}}\right)$$

Optimizing the captioning loss encourages the ECG features to accurately predict the tokenized text $t$ using an auto-regressive mechanism, conditioned on the ECG tokens $e$ generated by the ECG encoder and the Multimodal Text Decoder parameters $\theta$:

$$\mathcal{L}_{captioning} = -\sum_{l=1}^{L} \log P_\theta(t_l | t_{<l}, e)$$

## 4.2 SPATIAL-TEMPORAL EMBEDDING

In addition to the Transformer's positional embedding $PE = \{E_1, E_2, \ldots, E_N\}$, we incorporate the Spatial-Temporal (ST) embedding proposed by Jin et al. (2025) to enhance the model's ability to capture lead-specific and temporal information for each ECG token, as illustrated in Figure 4. For spatial embedding, given the input is a 12-lead ECG signal, we define a set $SE = \{E_I, E_{II}, \ldots, V6\}$, assigning a unique embedding to each corresponding lead. For temporal embedding, we use $TE = \{E_{t_1}, E_{t_2}, \ldots, E_{t_P}\}$, where $P$ denotes the number of patches obtained after splitting the original 10-second signal.

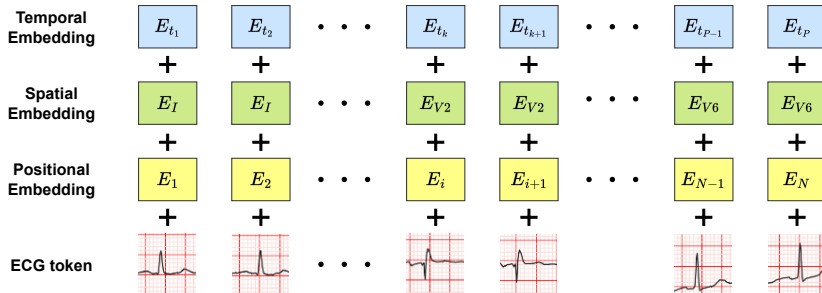

Figure 4: Prior to being processed by the ECG encoder, each ECG token is enriched with Spatial and Temporal embeddings in addition to standard Positional embeddings. This augmentation provides the model with explicit information about the lead identity and time frame associated with each token, enhancing its ability to capture spatial-temporal patterns in the ECG signal.

## 4.3 ATTENTION-GUIDING FOR EXPLAINABLE MODEL

To enable the model to learn which regions to attend to, we leverage the Transformer architecture Vaswani et al. (2023) and its inherent attention mechanism. Specifically, we introduce an Attention-Guiding procedure that incorporates an auxiliary loss function targeting the attention scores. To preserve the model's ability to initially consider all components, the input is processed in the standard manner. However, during training, the attention scores from the final Transformer layer are regularized using a Kullback–Leibler (KL) divergence loss, formulated as follows:

$$\mathcal{L}_{attention} = \frac{1}{B} \sum_{i=1}^{B} \sum_{n=1}^{N} p_{i,n} \frac{p_{i,n}}{a_{i,n}^{CLS}},$$

where $a_{i,n}^{CLS}$ denotes the attention score between the CLS token and the $n$-th token of the $i$-th sample, while $p_{i,n}$ represents the constructed target attention heatmap for token $n$ of sample $i$. The variables $B$, $N$ refer to the batch size and the total number of ECG tokens, respectively. Since the target heatmap highlights anomalous regions, this mechanism guides the model to focus on diagnostically relevant areas, thereby improving its decision-making capability. For samples exhibiting normal heart conditions, where no abnormal heatmap can be constructed, the attention loss $\mathcal{L}_{attention}$ is set to zero. The overall training loss is defined as:

$$\mathcal{L}_{total} = \alpha \mathcal{L}_{contrastive} + \beta \mathcal{L}_{captioning} + \gamma \mathcal{L}_{attention}$$

where $\alpha$, $\beta$ and $\sigma$ are hyperparameter weights corresponding to the contrastive, captioning, and attention-guiding loss terms, respectively.

## 5 Experiment

### 5.1 Dataset

To ensure a fair comparison with baseline models, we pretrain our model using the MIMIC-IV-ECG dataset Gow et al. (2023), which comprises over 800,000 12-lead ECG recordings of 10-second duration from approximately 160,000 unique patients. During preprocessing, any "NaN" or "Inf" values in the ECG signals are replaced with zero. For attention-guided supervision, we incorporate the synthesized `MIMIC-IV-ECG+X` as an auxiliary training heatmap for `X-ECG`.

For evaluation, we employ the PTB-XL dataset Strodthoff et al. (2020), a clinical 12-lead ECG dataset containing 21,837 recordings from 18,885 patients. This dataset is used for both arrhythmia classification and the generation of `PTB-XL+X`. We benchmark the classification task across four data variants: Diagnostic, Rhythm, Form, and All. For the report generation task, we adopt the LLM pretraining, fine-tuning, and benchmarking protocols established by ECG-Chat Zhao et al. (2025).

### 5.2 Configuration

During the pretraining phase, we utilize a 1D 12-layer Vision Transformer with a patch size of 50 as the ECG encoder. The text encoder is initialized with the pretrained MedCPT model Jin et al. (2023) and remains frozen throughout this process. To ensure that the CLS token attends to all input tokens, we first flatten the 12-lead ECG signal before feeding it into the encoder.

The model is trained with a learning rate of $1e - 4$ over 20 epochs, with the first 10,000 steps designated for warm-up. Training is conducted using a batch size of 64 across $4\times$ NVIDIA A100 GPUs, each with 80 GB of memory. For the loss weights, we adopt $\alpha = 1$ and $\beta = 2$, following the configuration in Zhao et al. (2025). The attention-guiding weight $\gamma$ is set to 0.5 to prevent it from exerting excessive influence during the training of `X-ECG`.

### 5.3 Metrics

**Arrhythmia Classification.** We adopt the FMax score Strodthoff et al. (2020), Area Under the Precision-Recall Curve (AUPRC), and Area Under the Receiver Operating Characteristic Curve (AUROC) as evaluation metrics for this classification task. To ensure balanced assessment across all diagnostic categories, we report the macro-averaged versions of these metrics, which compute the mean performance across all classes regardless of class imbalance.

**Anomaly localization.** To evaluate whether the attention scores of `X-ECG` accurately highlight relevant regions within each data sample, we frame this as a binary segmentation task. For assessment, we employ instance-averaging metrics including AUPRC and AUROC.

**Report generation.** We evaluate the output of the LLM conditioned on ECG representations using BLEU-2, BLEU-4, ROUGE-L, and METEOR to assess lexical similarity between the generated and reference reports—both at the word level and across contiguous word sequences. Additionally, we employ BERT-score to assess semantic similarity, capturing how closely the generated report aligns with the meaning of the reference report.

## 6 Results and Analysis

### 6.1 Arrhythmia classification

Upon completing the pretraining phase of `X-ECG`, we attach a linear classification head to the architecture. This layer utilizes the CLS token embedding as input. To demonstrate the quality of the learned representations, we freeze the ECG encoder and train only the linear head. For baseline comparisons, we utilize the official pretrained weights and apply the same procedure. Table 2 presents the classification results across multiple PTB-XL tasks Strodthoff et al. (2020).

Overall, `X-ECG` significantly outperforms all baseline methods across the four classification tasks—Diagnostic, Rhythm, Form, and All—on every evaluation metric. This highlights the model's strong ability to capture clinically relevant features from ECG input signals, resulting in more ac-

Table 2: Linear probing classification result evaluated on PTB-XL dataset. **Bold** indicates the best result, and underline indicates the second-best result

| Task | Methods | Explainability | FMax | AUPRC | AUROC |
|---|---|---|---|---|---|
| *Diagnostic* | ST-MEM Na et al. (2024) | ✗ | 27.24 | 21.55 | 86.20 |
| | MERL$_{Resnet}$ Liu et al. (2024) | ✗ | 25.88 | 19.46 | 82.92 |
| | MERL$_{ViT}$ Liu et al. (2024) | ✗ | 29.99 | 24.64 | 86.43 |
| | ECG-FM McKeen et al. (2025) | ✗ | 30.42 | 24.58 | 88.07 |
| | X-ECG (Ours) | ✓ | **41.34** | **34.72** | **92.01** |
| *Rhythm* | ST-MEM Na et al. (2024) | ✗ | 56.64 | 49.18 | 95.32 |
| | MERL$_{Resnet}$ Liu et al. (2024) | ✗ | 40.87 | 35.43 | 85.61 |
| | MERL$_{ViT}$ Liu et al. (2024) | ✗ | 27.98 | 21.43 | 74.44 |
| | ECG-FM McKeen et al. (2025) | ✗ | 52.97 | 46.11 | 85.37 |
| | X-ECG (Ours) | ✓ | **60.72** | **55.46** | **95.88** |
| *Form* | ST-MEM Na et al. (2024) | ✗ | 33.21 | 27.10 | 81.57 |
| | MERL$_{Resnet}$ Liu et al. (2024) | ✗ | 23.47 | 17.61 | 68.79 |
| | MERL$_{ViT}$ Liu et al. (2024) | ✗ | 27.98 | 21.43 | 74.44 |
| | ECG-FM McKeen et al. (2025) | ✗ | 27.83 | 20.57 | 78.74 |
| | X-ECG (Ours) | ✓ | **38.85** | **31.87** | **86.24** |
| *All* | ST-MEM Na et al. (2024) | ✗ | 30.79 | 24.89 | 87.16 |
| | MERL$_{Resnet}$ Liu et al. (2024) | ✗ | 26.01 | 19.61 | 81.01 |
| | MERL$_{ViT}$ Liu et al. (2024) | ✗ | 29.58 | 24.02 | 83.87 |
| | ECG-FM McKeen et al. (2025) | ✗ | 31.07 | 24.26 | 85.20 |
| | X-ECG (Ours) | ✓ | **41.68** | **34.76** | **91.50** |

Table 3: Anomaly Localization result evaluated on `PTB-XL+X`. **Bold** indicates the best result, and underline indicates the second-best result

| Methods | AUPRC | AUROC |
|---|---|---|
| ECGAD Jiang et al. (2023a) | 33.59 | 65.56 |
| ECG-Chat Zhao et al. (2025) | 18.37 | 50.57 |
| X-ECG (Ours) | **43.78** | **80.05** |

curate diagnostic predictions. Notably, X-ECG is the only approach that explicitly incorporates explainability, a crucial attribute for real-world clinical deployment. These results validate the effectiveness of the proposed model architecture and training strategy, particularly in leveraging attention mechanisms and pretrained ECG representations.

## 6.2 ANOMALY LOCALIZATION

For the evaluation of our model attention score, to see what components contribute to the final decision output, we perform inference on the PTB-XL test set and benchmark it as a binary segmentation task, as shown in Table 3. To obtain the ground truth label for comparison, we use the same two-stage mechanism as before to generate the desired abnormal criteria heatmap in `PTB-XL+X`.

The attention scores produced by ECG-Chat exhibit limited reliability, as indicated by an AUROC value close to 50%, suggesting that its focus regions are nearly random. In contrast, our model, X-ECG, achieves the highest scores in both AUPRC and AUROC—even without exposure to any sample signals from the PTB-XL dataset—demonstrating strong generalization and robust representation learning. Interestingly, although ECGAD is specifically designed for anomaly localization, it only achieves the second-best performance on both metrics. As shown in Table 2 and Table 3, X-ECG is capable of performing both arrhythmia classification and anomaly localization, whereas ECGAD is limited to identifying abnormal regions without the ability to classify the underlying diagnosis.

Table 4: Report generation evaluation using the English-translated PTB-XL report as label

| Methods | BLEU-2 | BLEU-4 | ROUGE-L | METEOR | BERT-score |
|---|---|---|---|---|---|
| ECG-Chat Zhao et al. (2025) | 21.18 | 11.96 | 33.83 | 32.97 | 89.00 |
| X-ECG (Ours) | **21.46** | **12.28** | **34.19** | **33.07** | **89.08** |

### 6.3 REPORT GENERATION

We extend the LLaVA framework Liu et al. (2023), which was originally proposed as an end-to-end trained large multimodal model aligning a vision encoder with an LLM for joint visual-language understanding, to the ECG domain by connecting a pretrained ECG encoder with an LLM for automated report generation. To enable cross-modal alignment, a learnable projection layer is used to transform ECG embeddings into the text embedding space of the LLM. During this stage, both the ECG encoder and the LLM are kept frozen to preserve their pretrained knowledge, while the projection layer is optimized to establish an effective mapping between modalities. Once the projection layer is trained, we proceed with continual fine-tuning of the entire pipeline. In this stage, the ECG encoder remains unchanged, while the feed-forward layers of the LLM are updated using the LoRA framework Hu et al. (2021), and the projection layer is updated concurrently.

Table 4 shows a comparison of report generation quality between our model and the baseline Zhao et al. (2025), measured against the ground truth report. Our model consistently outperforms the baseline across all metrics, including BLEU-2, BLEU-4, ROUGE-L, METEOR, and BERT-score. These improvements reflect enhanced lexical precision, structural coherence, and semantic fidelity in the generated reports.

## 7 CONCLUSION

This paper presents X-ECG, the first *explainable* ECG foundation model along with two abnormal heatmap locations dataset MIMIC-IV-ECG+X and PTB-XL+X. With careful design of the architecture and loss function, X-ECG not only outperform all baselines that don't have explainability in both arrhythmia classification and report generation task across various metrics. The explainability of our model is represented by a heatmap that provides detailed, wave-level segmentation of anomalies in the ECG waveforms. Our generated heatmaps are consistent with manual inspection by a licensed cardiac specialist. The contribution of our paper also includes the release of wave-level annotations of anomalies in ECG waveforms on the largest published ECG database, facilitating further development of explainability paradigm in ECG research.

### LIMITATION

The data curation process for MIMIC-IV-ECG+X and PTB-XL+X relies on external clinical knowledge to identify abnormal conditions. However, this approach may overlook certain cases due to dependencies on patient-specific factors such as gender and age. As illustrated in Figure 6, nearly half of the heatmaps in the test set are only partially correct, indicating limitations in the current annotation strategy. To improve the accuracy and reliability of abnormal region identification, a more comprehensive algorithm could be adopted for enhanced heatmap generation. Additionally, the attention-guiding mechanism currently utilizes KL divergence as its loss function. However, alternative formulations such as cross-entropy could be employed to enforce stricter alignment between predicted attention distributions and ground truth.

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

# A   DEMONSTRATION OF `PTB-XL+X`

Figure 5 showcases representative examples of the generated heatmaps alongside their corresponding original text reports from `PTB-XL+X`.

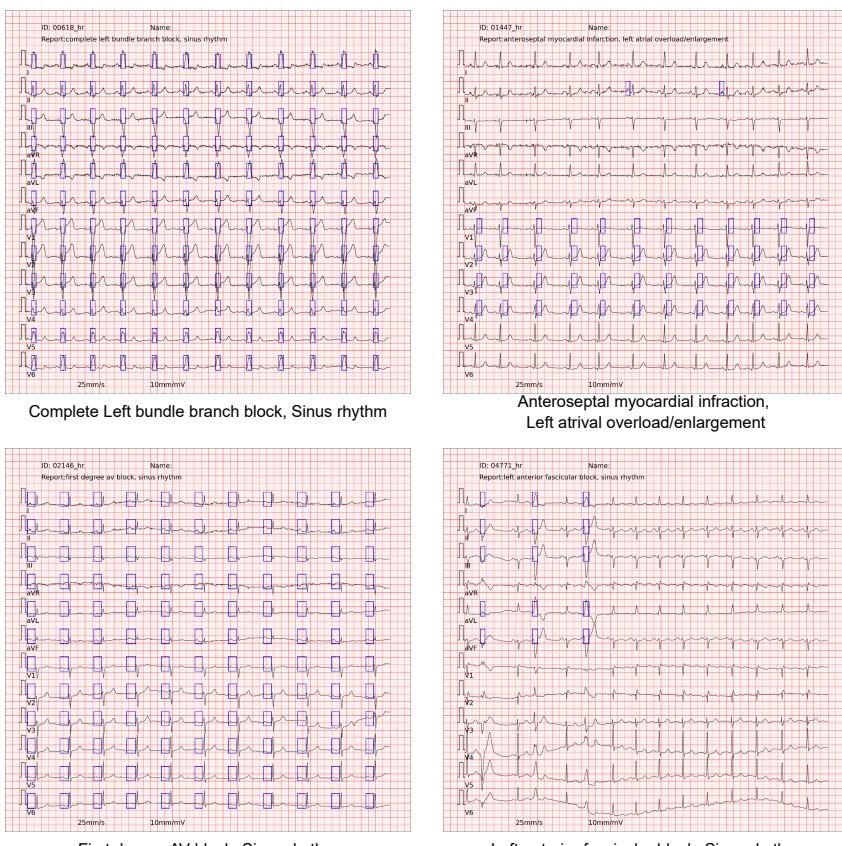

Figure 5: Example outputs from the two-stage rule-based procedure applied to the PTB-XL dataset. Blue boxes highlight the label regions of detected anomalies identified by our algorithm.

To statistically evaluate the quality of our dataset, we submitted 100 samples to cardiologists for expert review. The assessment focused on verifying the accuracy of the abnormal heatmap annotations. The results of this evaluation are presented in Figure 6.

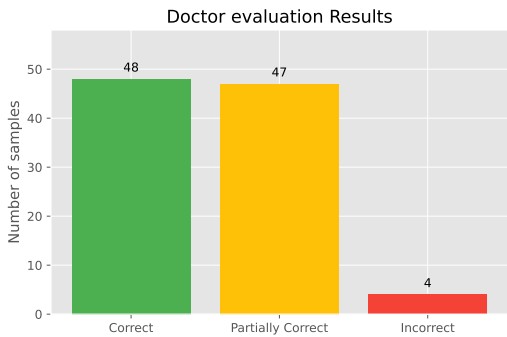

Figure 6: Evaluation results of abnormal heatmaps from 100 samples in `PTB-XL+X`, reviewed by a clinical expert

## B    DETAILS OF LARGE LANGUAGE MODELS USAGE

The use of Large Language Models (LLMs) in this work is limited solely to grammar correction and stylistic refinement. All core aspects—including the formulation of main contributions, experimental design, and data analysis—were conducted independently without LLM involvement.

