# OpenReview forum: "X-ECG: Explainable Foundation model for Electrocardiogram"
_ICLR.cc/2026/Conference — ICLR 2026 Conference Withdrawn Submission_

### Official Review · Reviewer_Mwur · 2025-10-30

**Soundness:** 2
**Presentation:** 2
**Contribution:** 2
**Rating:** 2
**Confidence:** 4

**Summary:**

**Summary**

X-ECG presents the first explainable ECG foundation model that combines wave-level anomaly detection with attention-guided pretraining. The authors curate MIMIC-IV-ECG+X and PTB-XL+X datasets using a two-stage rule-based annotation process, then train a model with multimodal alignment and attention regularization via KL divergence. The method achieves strong performance on arrhythmia classification, anomaly localization, and report generation tasks across multiple datasets.

**Strengths:**

- **Clinical relevance is clear and well-motivated.** Addresses a real clinical need for interpretable cardiology models, aligning with domain requirements for transparent, clinician-facing explanations.



- **Annotation pipeline is systematic and knowledge-grounded.** A two-stage rule-based process—abnormality seeking followed by criteria seeking—ties supervision to explicit clinical rules, improving traceability and auditability.



- **Evaluation is comprehensive across tasks and metrics.** Benchmarks cover classification, localization, and report generation with AUROC, AUPRC, BLEU, ROUGE, and BERTScore, demonstrating broad applicability.



- **Spatial-temporal embedding is well designed for ECG structure.** Explicit encoding of lead identity and temporal information leverages the multi-lead, sequential nature of ECGs to support finer-grained localization and robust representation.



- **Expert validation reinforces clinical orientation.** A clinical expert review of 100 samples, despite imperfect agreement, evidences a concrete commitment to assessing annotation quality and practical relevance.

**Weaknesses:**

- **Brittle rule-based supervision undermines reliability.** The framework’s core supervision comes from heuristic, rule-generated heatmaps; the paper’s own analysis (e.g., Figure 6) reports ~47% “Partially Correct,” acknowledging notable imperfections. Training with KL divergence to exactly match these noisy targets risks hard-coding errors and biases, weakening both robustness and the claim of faithful explainability.
- **Core contribution is unproven without** \mathcal{L}_{\text{attention}} ablation. The paper shows that X-ECG (full) outperforms baselines, but omits the decisive control: X-ECG without \mathcal{L}_{\text{attention}}. Without this, gains could stem from the stronger backbone (e.g., CoCa + spatial–temporal embeddings) rather than the proposed attention-guiding loss, leaving the central claim unsubstantiated.
- **Localization evaluation is circular.** Anomaly localization (Table 3) is benchmarked on PTB-XL+X, a set produced by the same two-stage rule pipeline used for training label generation. This setup primarily tests imitation of the rule-maker, not generalization to independently annotated or clinically adjudicated ground truth, so high scores do not translate to clinical validity.
- **Key hyperparameters and loss choice lack justification.** The attention weight is fixed at \gamma=0.5 “to prevent excessive influence,” with no sensitivity analysis; KL is chosen over alternative distribution-matching losses (e.g., cross-entropy) without empirical comparison. Absent robustness sweeps, the method’s stability and optimality remain in doubt.
- **Architectural attributions are unsubstantiated.** The paper asserts that spatial–temporal (ST) embeddings “significantly improve” localization, yet provides no ablation (e.g., X-ECG without ST) to quantify their contribution. Without isolating architectural effects from objective design, the source of observed gains remains ambiguous.

**Questions:**

1.	What is the correlation between heatmap quality (as assessed by cardiologist) and downstream task performance? A quantitative analysis would strengthen the paper.



2.	How sensitive is the method to the attention-guiding weight $\gamma$ in the attention loss?



3.	Can you provide failure case analysis where the model’s attention is clinically incorrect? What patterns lead to such failures?



4.	How does the method perform on rare conditions where rule-based annotations might be incomplete?

---

### Official Review · Reviewer_Pcbo · 2025-10-30

**Soundness:** 2
**Presentation:** 3
**Contribution:** 2
**Rating:** 2
**Confidence:** 5

**Summary:**

This paper presents X-ECG, the first explainable foundation model for electrocardiogram (ECG) analysis. The authors construct two annotated datasets—MIMIC-IV-ECG+X and PTB-XL+X—by automatically labeling abnormal ECG waves, intervals, and segments using clinically validated rules. Based on these datasets, X-ECG integrates an attention-guided learning framework that aligns model attention with pathological regions, enhancing both interpretability and diagnostic accuracy. Built upon a multimodal contrastive–captioning architecture with spatial–temporal embeddings, X-ECG achieves state-of-the-art results in arrhythmia classification, anomaly localization, and automated report generation, while providing fine-grained, clinically meaningful explanations.

**Strengths:**

- The manuscript is clearly written and easy to follow, making it accessible and straightforward to understand.
- The authors constructed two wave-level annotated datasets, MIMIC-IV-ECG+X and PTB-XL+X, which fill an existing gap in the field by providing fine-grained abnormality annotations for ECG signals.

**Weaknesses:**

- Lack of novelty. Most components of this manuscript—such as the Criteria Feature Retrieval (CFR) module, the ECG-CoCa architecture, and the spatial-temporal embedding—are directly derived from previous works, with no substantial methodological innovation presented.
- Limited advancement in explainability. The paper does not demonstrate a significant breakthrough in interpretability. Applying explainable AI techniques to ECG analysis has already been extensively explored (e.g., [1]). Although the paper claims to introduce the first explainable ECG foundation model, its core explainability mechanism — the attention-guiding loss aligned with rule-based heatmaps — still belongs to the conventional attention-map-based explainability paradigm. Attention weights only reflect the correlation between input regions and the model’s output, rather than causal relationships or diagnostic reasoning. The model may “look” at the correct regions, but it still lacks explicit reasoning or human-interpretable logic that connects waveform morphology to diagnostic conclusions. In my view, this does not constitute genuine explainability.
- Potential bias and dataset reliability concerns. While the paper introduces two valuable datasets, MIMIC-IV-ECG+X and PTB-XL+X, the reliability of the heatmap annotations remains uncertain. The expert evaluations presented in the appendix are conducted after model training, meaning that the observed inaccuracies could stem either from the rule-based annotation process or from biases introduced during model optimization. Nearly half of the evaluated samples are reported as only partially correct, which raises concerns about the robustness of both the dataset and the model’s learned attention mechanism. If the annotations cannot achieve high accuracy even under in-domain testing, it is difficult to trust their interpretability or generalizability in real-world out-of-domain scenarios.
- Incomplete baseline comparison. The manuscript partially adopts techniques from TolerantECG [2], yet the final experiments do not include TolerantECG as a baseline. Including it in the comparison would ensure fairness and clarify the incremental contribution of X-ECG relative to existing approaches.
- Lack of code availability. The authors appear unwilling to release their code, which raises concerns regarding the reproducibility and verifiability of the reported results.

[1] Hempel, P., Ribeiro, A.H., Vollmer, M. et al. Explainable AI associates ECG aging effects with increased cardiovascular risk in a longitudinal population study. npj Digit. Med. 8, 25 (2025). https://doi.org/10.1038/s41746-024-01428-7

[2] Huynh Dang Nguyen, Trong-Thang Pham, Ngan Le, and Van Nguyen. 2025. TolerantECG: A Foundation Model for Imperfect Electrocardiogram. In Proceedings of the 33rd ACM International Conference on Multimedia (MM '25). Association for Computing Machinery, New York, NY, USA, 8097–8105. https://doi.org/10.1145/3746027.3755287

**Questions:**

None

---

### Official Review · Reviewer_VQkY · 2025-11-02

**Soundness:** 3
**Presentation:** 3
**Contribution:** 2
**Rating:** 4
**Confidence:** 3

**Summary:**

X-ECG is a foundation model that targets  built-in explainability by steering model
attention toward regions flagged as abnormal by rule-based clinical heuristics and a trained
ECG wave segment technique. The authors employ public datasets, auto-generate wave/interval
annotations, and train a fairly standard ECG-text model with an auxiliary attention loss. On
PTB-XL/MIMIC benchmarks, they report gains in classification, localization, and
report generation compared to prior systems, and claim to be the first explainable ECG FM. In
essence: the proposal is a pipeline that puts together heuristic annotations with attention guidance to get better
performance and produce plausible visual highlights.

**Strengths:**

The paper targets a relevant and challenge application. Technically, it’s main strength is an integration of explainability into pretraining: instead of treating attribution as a post-hoc task, the authors use clinically motivated supervision while learning the signal, a creative combination of established data (ECG-text, contrastive/captioning objectives, wave segmentation, rule-based heuristics) that creates a
method which represents a contribution, although not a major one. Regarding quality, the experimental assessment is quite broad: classification, localization, and report generation showing consistent gains against relevant baselines on widely used public datasets, with a training recipe that seems to be reproducible. In terms of clarity, the paper states clearly that attention should be guided by clinically salient regions, and the end-to-end narrative, from curation to supervision to downstream evaluation. It reads well. With respect to significance, the work proposes a practical strategy for explainable biosignal FMs.

**Weaknesses:**

The main weakness of the paper is regarding its limited contribution, since it stitches together public ECG datasets, off-the-shelf wave segmentation, and rule-based heuristics with a fairly standard ECG-text training. The claimed “first explainable ECG FM” seems to be overstated. The explainability is plausible but it is not clear to what extent it is faithful, once the model is trained to match its own heuristic masks, inviting circularity, and there are no causal tests (deletion/insertion, AOPC, counterfactuals) to show that the explanations highlight truly drive predictions. Important experimental assessment dimensions are missing, so that the paper does not provide much about the actual contribution of attention guidance, segmenter/CFR, or temporal embeddings. Reported gains are not statistically significant (no multi-seed variance, confidence intervals, or significance tests; no calibration/robustness analysis). The human study seems to be quite limited (apparently a single rater, ~100 cases, no inter-rater agreement), and there’s an inconsistency in Figure 6 (48 + 47 + 4 does not sum to 100).

**Questions:**

How many cardiologists participated in the human study?

Did you compute inter-rater reliability?

The figure/text suggests a total of 100 cases but bar counts does not sum to 100.

---

### Official Review · Reviewer_8nkU · 2025-11-05

**Soundness:** 3
**Presentation:** 2
**Contribution:** 2
**Rating:** 2
**Confidence:** 3

**Summary:**

The paper present an explainable ECG foundation model based on an attention-guided training process. By curating the public datasets with Abnormal seeking and Criteria seeking, the authors train a model to align ECG signals and text and also enables it to highlight the relevant regions. It shows good performance on three tasks: arrhythmia classification, report generation and anomaly localization.

**Strengths:**

1. The techniques are sound. It is intuitive that the ECG signals are not relevant to all texts, and attending to focused regions should help a lot. The paper solved this problem in a reasonable way and showed good performance on classification and generation tasks.
2. The biggest contribution and technical difficulty is the data curation. If the data can be public, I think it can improve many multimodal ECG models.

**Weaknesses:**

The overall architecture for the machine learning model is not novel, making the technical contribution a little bit trivial. Most of the modules are from previous works, e.g. the Spatial-Temporal (ST) embedding is from Jin et al. 2025, and attention-mechanisms are largely used in previous ECG models.
As I mentioned above, the main contribution of this paper seems to be the data curation part, instead of the modeling part.

**Questions:**

As far as I understand, the patch size of the ECG tokens is fixed at 50, but the anomaly may occur at different resolutions of the ECG signals. Does the patch size impact the performance a lot?

---

### Note · Authors · 2025-12-01

I have read and agree with the venue's withdrawal policy on behalf of myself and my co-authors.